# Supplementary Information

## 5 Dynamical Similarity

We consider the problem of determining when two linear autonomous dynamical systems on $\mathbb{R}^n$

$$\dot{\mathbf{x}} = \mathbf{A}\mathbf{x} \quad \text{and} \quad \dot{\mathbf{y}} = \mathbf{B}\mathbf{y} \tag{9}$$

may be regarded as equivalent. One natural definition is that for all time $t$, there exists a linear relationship between $\mathbf{x}$ and $\mathbf{y}$

$$\mathbf{y} = \mathbf{C}\mathbf{x} \tag{10}$$

Differentiating both sides with respect to time and substituting in the dynamics relationship yields an equivalent condition in terms of the system matrices $\mathbf{A}$ and $\mathbf{B}$

$$\mathbf{y} = \mathbf{C}\mathbf{x} \iff \mathbf{B} = \mathbf{C}\mathbf{A}\mathbf{C}^{-1} \tag{11}$$

Thus a natural measure of *similarity* between two linear dynamical systems is

$$\mathrm{d}(\mathbf{x}, \mathbf{y}) \equiv \min_{\mathbf{C}} ||\mathbf{B} - \mathbf{C}\mathbf{A}\mathbf{C}^{-1}|| \tag{12}$$

### 5.1 Computing Procrustes Alignment over Vector Fields

Unfortunately, this measure of similarity requires solving a nonconvex optimization problem. To simplify, we restrict $\mathbf{C}$ to be orthogonal, i.e., $\mathbf{C}^{-1} = \mathbf{C}^T$ or $\mathbf{C} \in O(n)$, the orthogonal group. The problem is still nonconvex, but we can take advantage of a convenient parameterization of orthogonal matrices in terms of the *Cayley Transform*. Namely, for an arbitrary skew-symmetric matrix $\mathbf{S}$, the matrix

$$\mathbf{C} = (\mathbf{I} - \mathbf{S})(\mathbf{I} + \mathbf{S})^{-1} \tag{13}$$

is orthogonal and may readily be computed in deep learning libraries such as PyTorch (e.g., `C = torch.linalg.solve(I + S, I - S)`). Using this parametrization, we compute the metric with gradient descent over $\mathbf{S}$ via the Adam optimizer.

The Cayley Transform restricts the operation to the special orthogonal group $SO(n)$, which is the group of orthogonal matrices that does not include reflections. We cannot continuously parametrize all of $O(n)$, as it is made up of two disconnected components: $SO(n)$, where for all $C \in SO(n)$, $\det(C) = 1$, and $O(n)/SO(n)$, where determinants = -1. Note that $SO(n)$ is a subgroup of $O(n)$ while $O(n)/SO(n)$ is not, as $O(n)/SO(n)$ does not have the identity element. To optimize over each with gradient descent, we first state a theorem:

**Theorem 1.** *All elements of $O(n)/SO(n)$ are bijective maps between itself and $SO(n)$.*

*Proof.* Let $D, P \in O(n)/SO(n)$. Then $PD \in SO(n)$. For this to be true, $PD$ must have two properties: $\det(PD) = 1$, and $(PD)^T(PD) = I$. The first is easily shown: $\det(PD) = \det(P)\det(D) = -1 * -1 = 1$. The second is likewise shown easily: $(PD)^T(PD) = D^T P^T PD = D^T D = I$ by orthogonality. Both $P$ and $D$ have inverses $P^T$ and $D^T$.

Conversely, let $D \in SO(n)$. Then we can show by the same logic that $PD \in O(n)/SO(n)$. $\quad \square$

From this theorem, we see that we can optimize over $O(n)/SO(n)$ by fixing some matrix in this subset (for example, a permutation matrix that swaps two indices). Then alongside the original formulation of the metric, we can optimize over

$$\min_{C \in SO(n)} ||B - CPAP^T C^T||_F \tag{14}$$

and take the minimum of the two values to be our final score.

# 6 The minimum similarity transform is a metric

Let A and B each be matrices in $\mathcal{R}^{n \times n}$. We wish to show that

$$d(A, B) = \min_{C \in O(n)} ||A - CBC^{-1}||_F \tag{15}$$

is a proper metric–that is, the following three properties hold:

- Zero: $d(A, B) = 0 \iff A = B$
- Symmetry: $d(A, B) = d(B, A)$
- Triangle Inequality: $d(A, C) \leq d(A, B) + d(B, C)$

*Proof.* According to Williams et al. [2021], only two things are needed. First, we must show that

$$g(A, B) = ||A - B||_F \tag{16}$$

is a metric, which is trivially true. Then we must show that the similarity transform is an isometry:

$$g(T(A), T(B)) = g(A, B) \tag{17}$$

where $T(A)$ is a map that in our case is the similarity transform.

$$g(T(A), T(B)) = ||CAC^{-1} - CBC^{-1}||_F = ||C(A - B)C^{-1}||_F \tag{18}$$

Using the identity that the Frobenius norm of a matrix is equivalent to the trace of the inner product with itself,

$$g(T(A), T(B)) = \text{Tr}([C(A - B)C^{-1}]^T [C(A - B)C^{-1}]) \tag{19}$$

$$= \text{Tr}(C(A - B)^T C^{-1} C(A - B)C^{-1}) \tag{20}$$

$$= \text{Tr}(C(A - B)^T (A - B)C^{-1}) = \text{Tr}(C^{-1}C(A - B)^T (A - B)) \tag{21}$$

$$= \text{Tr}((A - B)^T (A - B)) = ||A - B||_F = g(A, B) \tag{22}$$

With the last steps using the cyclic property of the trace (Tr) and the fact that C is an orthonormal matrix. $\square$

# 7 The minimizer of the euclidean distance also minimizes angular distance

*Proof.* Let $C^*$ be the minimizer of $||A - CBC^{-1}||_F$. Then we can rewrite this as above, using the trace identity:

$$||A - CBC^{-1}|| = \text{Tr}((A - CBC^{-1})^T (A - CBC^{-1})) \tag{23}$$

which can be expanded into:

$$= \text{Tr}(A^T A) - 2\text{Tr}(A^T CBC^{-1}) + \text{Tr}(B^T B)$$

$$= \text{Tr}(A^T A) - 2 < A, CBC^{-1} > +\text{Tr}(B^T B)$$

By minimizing the Frobenius norm, $C^*$ must maximize the inner product $< A, CBC^{-1} >$. The angular distance is defined as

$$\arccos \frac{< A, CBC^{-1} >}{||A||||B||} \tag{24}$$

Due to the arccos function being montonically decreasing, maximizing the matrix inner product therefore minimizes the angular distance. $\square$

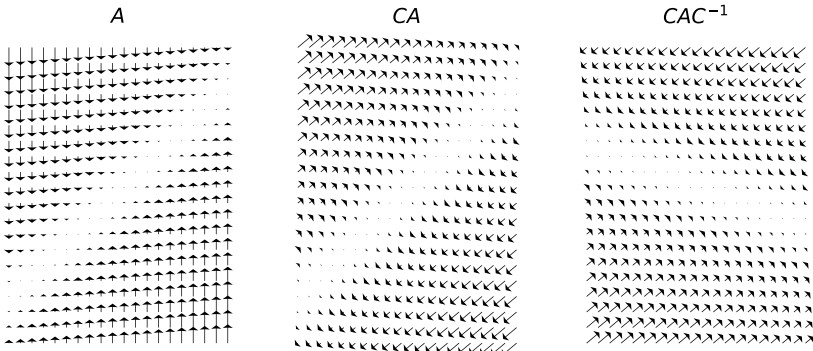

Figure 6: Vector Fields for a linear system $A$ and two transformations. Here, $C$ is an orthogonal matrix. The size of the arrow represents the magnitude of the vector, so the small dots are zeros, indicating the line attractor.

## 8  Vector Fields Under Orthogonal Transformation

Here we demonstrate how Vector Fields transform. Given some linear system $\dot{x} = Ax$, the matrix $A$ contains all the information about the vector field (Fig. 6,left). The transformation $\mathbf{CA}_y$ simply rotates vectors in place, which destroys dynamic structure (for example, it could turn a stable fixed point into an unstable one). However, $\mathbf{CA}_y\mathbf{C}^{-1}$ rotates their positions as well, thereby preserving the vector field structure and satisfying the same goal as $\mathbf{CY}$ applied to a data matrix.

Applying a single orthogonal matrix $C$ to the left (or right) of $A$ to yield the system $\dot{x} = CAx$ produces the plot in Fig. 6 middle. This is clearly a different dynamical system, as all the vectors now point away from the line attractor attractor, thereby making the system unstable instead of stable. However, applying the similarity transform $CAC^{-1}$ preserves the dynamical structure of the linear system by rotating both the vectors and their positions on the field (Fig 6, right).

## 9  Equivalency to 2-Wasserstein Distance

Here we prove a relationship between DSA and the 2-Wasserstein distance $W_2(X, Y)$. Because the equivalence of the eigenvalues in $A_x$ and $A_y$ is sufficient to show conjugacy of the two underlying systems (if they have point spectra, next section), the Wasserstein distance is also a useful metric to measure distance between two dynamical systems. This was utilized in Redman et al. [2023] to measure dynamical similarities between different learning algorithms.

**Theorem 2.** *DSA is equivalent to the 2-Wasserstein Distance between the eigenvalues of $A_x$ and $A_y$ when these matrices are normal.*

*Proof.*

$$DSA(A_x, A_y) = \min_{C \in O(n)} \left| A_x - CA_yC^T \right|_F \tag{25}$$

$$W_2(X, Y) = \inf_\pi \left( \frac{1}{n} \sum_{i=1}^n \left| X_{ii} - Y_{\pi(ii)} \right|_2 \right)^{1/2} = \min_{P \in \Pi(n)} \left| X - PYP^{-1} \right|_F \tag{26}$$

Where $X$ and $Y$ are sets represented as diagonal matrices. $\Pi(n)$ is the group of permutation matrices, a subgroup of the orthogonal group $O(n)$. When $A_x$ and $A_y$ are normal, they can be eigendecomposed as $A_x = V_x \Lambda V_x^T$ and $A_y = V_y \Delta V_y^T$. $\Lambda$ and $\Delta$ are diagonal matrices of eigenvalues. By the group properties of $O(n)$, $C$ can be written as the composition of other elements in the group. Let $C = V_x P V_y^T$ where $P \in O(n)$. Then $DSA$ is equivalent to

$$DSA(A_x, A_y) = \min_{P \in O(n)} \left| V_x \Lambda V_x^T - V_x P \Delta P^T V_x^T \right|_F = \min_{P \in O(n)} \left| \Lambda - P \Delta P^T \right|_F \tag{27}$$

Now we wish to show that the minimum occurs when P is a permutation matrix. Using the trace identity of the Frobenius norm, we can rewrite this as

$$DSA(\Lambda, \Delta) = \min_{P \in O(n)} \sqrt{\text{Tr}((\Lambda - P\Delta P^T)^T(\Lambda - P\Delta P^T))} \qquad (28)$$

which is minimized at the maximum of $\text{Tr}(\Lambda^T P \Delta P^T)$. In summation notation,

$$\max_{P \in O(n)} \text{Tr}(\Lambda^T P \Delta P^T) = \max_{\sum_j p_{ij}^2 = 1} \sum_i \Lambda_i \sum_j p_{ij}^2 \Delta_j = \max_{P \in O(n)} \text{Tr}(\Lambda P \circ P \Delta) \qquad (29)$$

$P \circ P$ is a doubly stochastic matrix, the set of which is a convex polytope whose vertices are permutation matrices (Birkhoff-von Neumann). $\text{Tr}(\Lambda P \circ P\Delta)$ is affine in $P \circ P$, which means its critical points are those vertices. Thus, $P \circ P$ and $P$ itself is a permutation matrix, $P \in \Pi(n)$. $\qquad \square$

## 10   Conjugacy and the Eigenspectra of the Koopman Operator

In our experiments, we demonstrated that DSA can identify conjugacy between different dynamical systems. But is this generally true, and do we have any theoretical guarantees on this? Here we recapitulate results from Koopman theory to show that DSA indeed has the properties necessary to identify conjugate dynamics.

First, recall that HAVOK and other DMD methods seek to identify a finite approximation of the Koopman operator from data. The Koopman operator is defined as the shift map of observables, $U_f^t \theta(x_0) = \theta(f^t(x_0))$, where our dynamical system is defined by the flow function $f^t$, with a time parameter that takes the state from $x_0$ to $x_t$. $\theta : X \to \mathbb{C}$ is function called an observable. Second, note that a similarity transform $CAC^{-1}$ preserves the eigenvalues and multiplicities of the matrix $A$. As seen above in Section 9, under some conditions DSA can be viewed as comparing the eigenvalues of the DMD matrices $A_x$ and $A_y$. Assuming that we are in the regime such that the DMDs converges to the Koopman approximation, the following result from Budišić et al. [2012] implies that DSA will report zero if the systems are conjugate.

**Theorem 3** (Spectral equivalence of topologically conjugate systems Budišić et al. [2012])**.** *Let $S : M \to M$ and $T :\to N \to N$ be maps that define two topologically conjugate dynamical systems over different manifolds $M$ and $N$. Let there exist a homeomorphism $h : N \to M$ such that $S \circ h = h \circ T$. If $\phi$ is an eigenfunction of the Koopman operator of $S$, $U_s$ (i.e. $U_s\phi = \lambda\phi$), then $\phi \circ h$ is an eigenfunction of the Koopman operator of $T$, $U_t$ also with eigenvalue $\lambda$.*

*Proof.* Define $x \in M$ and let $y \in N$ such that $x = h(y)$. Then

$$U_S\phi(x) = \phi S(x) = \lambda\phi(x)$$
$$\phi \circ (S \circ h)(y) = \lambda(\phi \circ h)(y)$$
$$\phi \circ (h \circ T)(y) = \lambda(\phi \circ h)(y)$$
$$U_T(\phi \circ h)(y) = \lambda(\phi \circ h)(y)$$

$\qquad \square$

We would also like to show the converse: if two Koopman operators have the same eigenspectra, their underlying systems are conjugate. We briefly sketch a proof here, which is captured in this commutative diagram. Here, $f$ and $g$ represent our nonlinear systems, with Koopman operators $U^f$ and $U^g$. $A_f$ and $A_g$ represent the valid linearizations:

$$
\begin{array}{ccc}
f & & g \\
\Big\uparrow{\scriptstyle U^f} & & \Big\uparrow{\scriptstyle U^g} \\
A_f & \xleftrightarrow{C^{-1}(\cdot)C} & A_g
\end{array}
$$

As seen in Lan and Mezić [2013], we can construct linearizations of nonlinear dynamical systems from Koopman eigenfunctions, that hold over an entire attractor basin. This corresponds to the vertical

arrows in the diagram. With these in hand, we can use a basic result in linear systems theory which also inspired the metric in DSA: Two linear dynamical systems are conjugate if their eigenvalues are equivalent, and the conjugate mapping is linear. Because these linear systems are one-to-one with the given nonlinear systems in a particular attractor basin, the diagram holds. Because each arrow is a conjugate mapping, the composition of three conjugacies is a conjugacy and thus the $f$ and $g$ are conjugate themselves.

## 11 DSA pseudocode

---
**Algorithm 1** Dynamic Similarity Analysis

---
**Input:** $X_1, X_2 \in \mathbb{R}^{n \times t \times d}, p, r$
**Output:** Similarity transform distance $d$ between the two dynamics matrices
    **procedure** DELAYEMBEDDING($X$, $p$)
        Initialize $H \in \mathbb{R}^{n \times (t-p) \times p*d}$
$$H = \begin{bmatrix} X[:, 1:p, :] \\ X[:, 2:p+1, :] \\ \cdots \\ X[:, t-p+1:t, :] \end{bmatrix} \text{ reshaped to } \mathbb{R}^{n \times (t-p) \times p*d}$$
        **return** H
    **end procedure**
    **procedure** DMD($H$,$r$)
        Initialize $A \in \mathbb{R}^{r \times r}$
        $H' = H[:, 1:]$
        $H = H[:, :-1]$
        Decompose $H = U\Sigma V^T$, and $H' = U'\Sigma'V'^T$
        Reduce Rank: $V = V[:, :r]$, and $V' = V'[:, :r]$
        Compute $A$ such that $V' = AV$ (OLS or Ridge Regression, depending on regularization)
        **return** A
    **end procedure**
    $H_1 = $ DELAYEMBEDDING($X_1$, $p$)
    $H_2 = $ DELAYEMBEDDING($X_2$,p)
    $A_1 = $ DMD($H_1$,r)
    $A_2 = $ DMD($H_2$,r)
    $d = \min_{\mathbf{C} \in \mathbf{O}(r)} ||\mathbf{A_1} - \mathbf{CA_2C}^{-1}||$

---

## 12 Eigen-Time Delay Coordinates is equivalent to PCA whitening.

In the Eigen-Time Delay Coordinate formulation of HAVOK, we compute the DMD over the right singular vectors of the SVD of H, $H = U\Sigma V^T$. PCA whitening normalizes the covariance matrix of the (already centered) data $H$:

$$HH^T = U\Sigma^2 U^{-1} \tag{30}$$

PCA whitening is equivalent to $H' \leftarrow \Sigma^{-1}U^{-1}H$. PCA whitening applied to the SVD of H is written as

$$H' \leftarrow \Sigma^{-1}U^{-1}U^{\Sigma}V^T = V^T \tag{31}$$

which is evidently the eigen-time delay formulation.

## 13 FlipFlop task MDS clustering by activation

Here, we plotted the same MDS scatterplots as in Fig. 2 in the main text. The only difference here is that the networks are colored by activation function instead of architecture. Evidently, the same conclusion holds here as in Fig. 2.

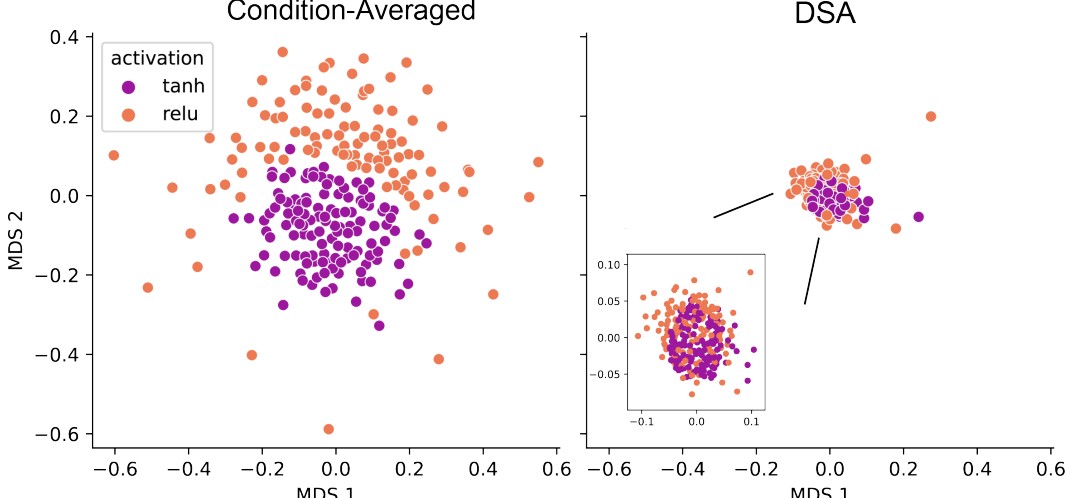

Figure 7: **Same dynamics, different shape, only identified as similar with DSA**. Dots indicate a single trained network. Analysis of RNNs trained on the 3-bit flipflop task. **b.** MDS Projection of the Dissimilarity Matrix computed across condition-averaged hidden states between each RNN (activation function indicated by color). In this view of the data, RNNs cluster by activation function. **c.** MDS Embedding of the dissimilarity matrix generated from DSA of the same set of RNNs as in b. Here, RNNs do not cluster by activation function.

# 14 Flip-Flop task HAVOK grid sweep

We fit HAVOK models with a range of ranks and delays, and assessed their fits in three different ways: $R^2$, MSE, and Pearson correlation between the predicted state from HAVOK and the true data, across 10 different models. Note that we did not exhaustively test all trained models and a large range of ranks and delays for the sake of efficiency. Here we see that across all metrics, steadily increasing rank and number of delays (listed as lag) improves the performance of the model. Across all metrics, there appears to be a Pareto Frontier which implies that the specific choice of rank and lag is not important. Furthermore, the metrics appear to saturate relatively quickly.

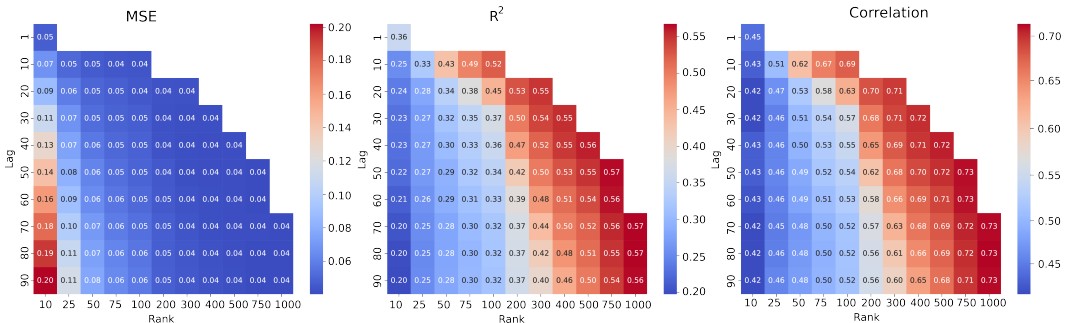

Figure 8: Mean HAVOK fits for three different scores across a variety of delays and ranks. The unfilled elements occurs when the rank is larger than the number of lags times the delays. Each model was computed over 15 Principal Components of the data, which had 128 trials with 500 timesteps each.

# 15 Condition-Averaged Aligned Dynamics Systems Equations

Here we describe the process we used to sample from the three dynamical systems with different attractors, and adversarially optimize input so that the condition averages are indistinguishable. This describes the systems tested in Figure 3 in the main text.

**Bistable System**

$$\dot{x}_1 = ax_1^3 + bx_1 + u + \epsilon_2 \tag{32}$$

$$\dot{x}_2 = cx_2 + u + \epsilon_2 \tag{33}$$

We randomly sample parameters for the system once from: $a \sim U(-5, -3)$ $b \sim U(4, 7)$ $c \sim U(-4, -2)$ and noise at each time step sampled i.i.d. $\epsilon \sim N(0, 0.05)$ We update the system with an Euler update with $dt = 0.01$. $u$ is defined as a constant in $\{-0.1, 0.1\}$ to drive the system to either of the stable fixed points.

**Line Attractor**   We define the eigenvectors and eigenvalues to get the linear system:

$$\Lambda = \begin{pmatrix} -1 & 0 \\ 0 & 0 \end{pmatrix} \quad V = \begin{pmatrix} 1 & 1 \\ 1 & 0 \end{pmatrix} \quad A = V\Lambda V^{-1} \tag{34}$$

$$\dot{x} = Ax + \hat{u} \tag{35}$$

**Point Attractor**

$$A = \begin{pmatrix} -0.5 & 0 \\ 0 & -1 \end{pmatrix} \tag{36}$$

$$\dot{x} = Ax + \hat{u} \tag{37}$$

**Adversarial optimization scheme to align condition averages**   We apply a simple feedback linearization scheme to control the condition averages of the line and point attractor. We simulate batches of trajectories simultaneously in order to calculate condition averages for the optimization scheme. First, we identify the condition averages $\bar{y}$ from the bistable system, which are previously calculated. At each step of the latter two systems, we calculate the condition averages $\bar{x}$. Then we define $\hat{u}$, the input that aligns the condition averages between systems:

$$\hat{u} = -A\bar{x}_t + \frac{1}{\alpha}(\bar{y}_{t+1} - \bar{x}_t) \tag{38}$$

Thus, when we apply the Euler update with time step $\alpha$, our system is defined by

$$x_{t+1} = x_t + \alpha(Ax_t - A\bar{x}_t) + \bar{y}_{t+1} - \bar{x}_t \tag{39}$$

Which makes our condition-averaged dynamics of the line and point attractor system

$$\bar{x}_{t+1} = \bar{y}_{t+1} \tag{40}$$

## 16   Ring Attractor Dynamics

We implemented the ring attractor defined in Wang and Kang [2022], specified by the following equation:

$$\tau \frac{ds}{dt} = -s + W^T \phi(s) + A \pm \gamma b(t) + \xi(x, t) \tag{41}$$

Here, $s$ describes the synaptic activation along the ring, and $\phi$ is the neural transfer function that converts these into firing rates: $r = \phi(s)$. In our simulations, $\phi$ was the ReLU function. To deform the geometry while preserving the ring topology, we applied the sigmoid transform in the main text to the extracted synaptic activations $s$ after simulation. To stabilize the dynamics of the ring for path integration, there are in fact two rings, each with their connectivity profile slightly offset, and with the $\pm$ term separately $+$ for one ring and $-$ for the other. $A$ describes the resting baseline input to the neurons, whereas $b$ describes the driving input that the network seeks to integrate, and $\gamma$ is a coupling term. $\tau$ describes the time constant, and $\xi$ describes the activity noise term. $W$ of course describes the weight matrix that has local excitatory connectivity and global inhibitory connectivity between neurons in the ring.

## 17 Line Attractor Deformation

We use the same line attractor matrix as in section 15:

$$\Lambda = \begin{pmatrix} -1 & 0 \\ 0 & 0 \end{pmatrix} \quad V = \begin{pmatrix} 1 & 1 \\ 1 & 0 \end{pmatrix} \quad A = V\Lambda V^{-1} \tag{42}$$

However, the dynamics are slightly different, as we introduce a nonlinear activation function:

$$\dot{x} = A \tanh(\beta x) + u + \epsilon \tag{43}$$

Where $u = \{1, -1\}$ and is constant across both dimensions to drive the system along the attractor in different directions. Here, $\beta$ is a scalar that affects the curvature of the attractor, as demonstrated by the vector fields of this system at three example values (Fig. 9a). We applied the same analysis as in Fig.4 in the main text and found the exact same results–Procrustes distance increases in magnitude as the two $\beta$ values being compared increases, but DSA is invariant. This adds to the evidence that DSA captures the topology of a dynamical system.

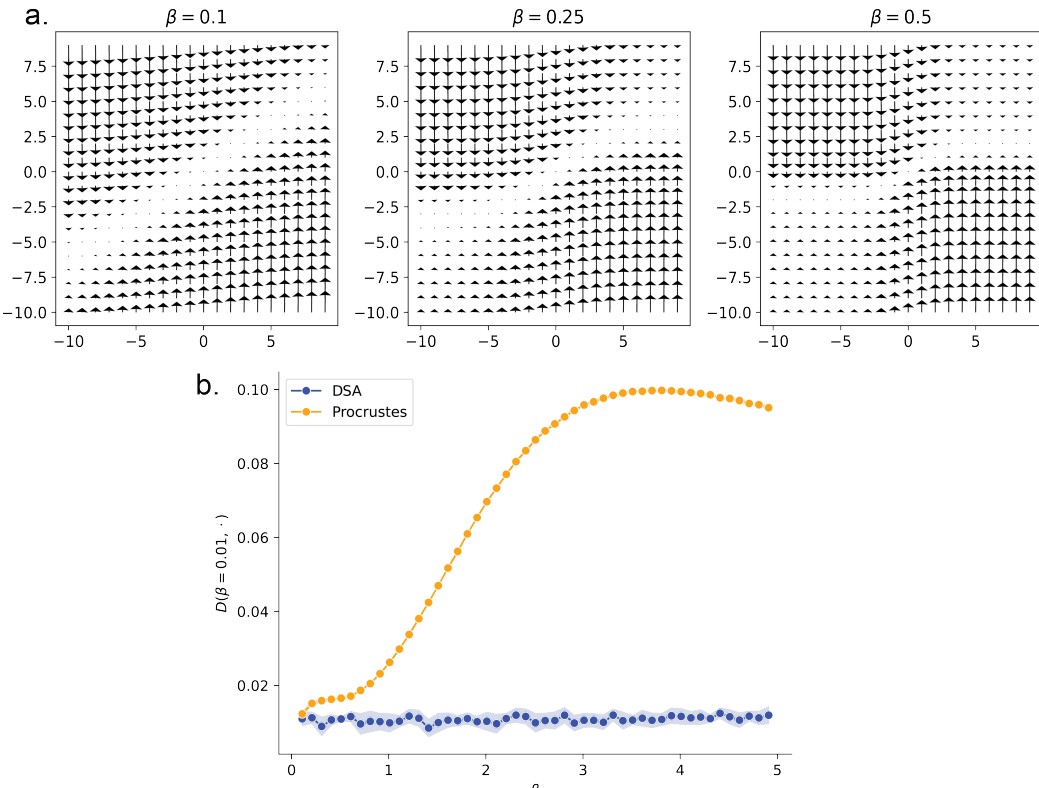

Figure 9: **DSA is invariant under geometric deformations when applied to a line attractor**. **a**. Demonstration of how the vector field transforms under $\beta$. Here, the line attractor becomes more s-shaped at higher $\beta$ values. **b**. Procrustes and DSA similarity to the smallest tested $\beta$ (0.1) at various values. Shaded region indicates standard error of the mean over 10 simulations.