# OpenReview forum: "Beyond Geometry: Comparing the Temporal Structure of Computation in Neural Circuits with Dynamical Similarity Analysis"
_NeurIPS.cc/2023/Conference — NeurIPS 2023 poster_

### Official Review · Reviewer_TLG4 · 2023-07-02

**Soundness:** 4 excellent
**Presentation:** 3 good
**Contribution:** 3 good
**Rating:** 7
**Confidence:** 4

**Summary:**

In this paper the authors propose a similarity metric to compare dynamical systems – in particular neural networks. They use a set of recent methods to transform a general dynamical system into a basis in which the dynamics is approximately linear. They compare two dynamical systems by comparing their linear dynamics in this transformed space by seeing how well one system can be turned into the other using only an orthogonal transformation. They test their metric on a variety of settings.

**Strengths:**

I think the idea is a good one.
I think it has been well executed.
The experiments were well chosen.
The writing was generally pretty good.

**Weaknesses:**

I don’t understand why section 2.3 was separate from section 3. It feels like section 2.3 played two roles – advertising the set of experiments that were going to test this new metric in interesting ways, and describing the experimental setup. The first of these is done in the first paragraph, and the second should, in my opinion, be moved into section 3. The separation between setup and results meant I had to keep flipping back, and you repeated yourself for no reason. It would make much more sense to me if each experiment was introduced and its results were discussed together.

I thought some of the experimental details were also unclear, and generally you rely too much on the reader recalling in detail many of the papers you cite. In particular:
1. I did not know what the 3-bit flip flop task was before reading this, and had to go find another paper that describes it. It seems crazy not to put a short description in the paper, even if it is in the appendix if page limits are the issue.
2. When trying to understand the ring attractor I eventually decided that r and s in equation 5 were phi(g) and g in equation 25. But then why change notation? And if I was right could this be made clear? If not, it was obviously not clear enough for this dumbo to understand what was going on.
3. Further, in appendix 9 you say “in our deformation analysis we only changed phi after simulation, which does not affect the ring topology but modifies the geometry”
I had no idea what this meant. In my head simulation is running the network dynamics, so you’re saying you only changed the non-linearity after you’d simulated, in which case it would have… zero effect. I must be misunderstanding?

The claim that DSA measures topological structure of the dynamics would be fairly remarkable and does seem to be backed up by the experiments. However it doesn’t seem at all proven – as you yourself point out in the discussion. In fact, what exactly is being compared through the dynamics of the eigenspace of these lagged timeseries is a little opaque to me – and I might wager to you too. Given this lack of certainty, I have two specific places I think the writing could use a fudge word, like in the discussion where you say “DSA _may_ be capturing similarity at the level of topology”. Namely:
1. Line 251: We demonstrate DSA compares two dynamical systems at the level of their topology.
2. Line 266: This suggests that DSA identifies similarity at the level of the topology…

Could you just write '_may identify similarity_', or '_is consistent with comparing at the level of topology_', because the pure claim “We demonstrate DSA compares two dynamical systems at the level of their topology” does not seem well founded.


**Questions:**

-	It seems like the scale of the MDS dimensions is being argued to be meaningful in your plots. Beyond some vague intuition that bigger means more variance, are the dimensions meaningful? What do their scales correspond to?
-	How well do the linearised dynamics capture the underlying network dynamics? In my limited knowledge of the HAVOK methodology (gathered from Steve Brunton’s excellent youtube videos) the system cannot usually be modelled as a linear system without including a forcing term. Do you find the same thing? If so do you not think it could be that key aspects of the dynamics are entering through the dynamics of the forcing term, slightly weakening the power of your method. Regardless, I think it would be good to persuade people that these approximate dynamics you are comparing are actually good approximations.
-	I did not buy one claim, could you elaborate:
“By linearity of PCA, the dynamics of the original system within the PC subspace are preserved” - Line 224.
Take the limit, say the dynamics of the system is just circling round an ellipse and you perform PCA to reduce that to 1D, the major axis of the ellipse. Sure, the dynamics on that 1D are the same as the original system but surely you lose something in this projection? And wouldn't that potentially be true of your original system too?
-	You claim that all networks that solve the 3-bit Flip-Flop task have the same 8 fixed point structure, line 219, do you have any evidence for this? Or is this from Maheswaranathan et al. 2019? If from Maheswaranathan, make it clear, cite them straight after you make the claim!
-	Why did you not choose figure 3 to also back up your claim that DSA might be capturing similarity at a topological level? (line 317)
-	Related to the topological claim, my intuition must be off. Humour me for a moment and ignore the time lagging: if you had activity rotating on some ring, then morphed the behaviour to become ellipsoidal, the topology would be the same, but the rotation matrices would be different. Further, the two matrices could not be mapped into one another by an orthogonal transformation. As such, in my intuition your method would (potentially very reasonably) say these were different dynamics even if the topology was the same. Given this, why do you think your method is going to care only about topology? Is it the time-lagging doing some magic?



**Limitations:**

The authors include a section of limitations that sound reasonable.

---

> ### Author Rebuttal · Authors · 2023-08-08
>
> We are thankful for the reviewer's extensive feedback and appreciate their comments--we believe we can answer all of their questions, which correspond to valuable improvements to the paper.
>
> > I don't understand why section 2.3 was separate from section 3...
>
> This is a reasonable point and we agree that we can restructure. Our intention was to follow the style of previous papers that leave technical descriptions of experiments in Methods alone. However, we agree that it would be more readable (and less redundant) to combine them. Thank you for the suggestion!
>
> > I did not know what the 3-bit flip flop task was before reading this...
>
> We greatly appreciate this feedback - it is important to ensure the reader is clear on the task structure so that our results can be fully understood. We will introduce the task in more detail, and can add another panel (similar to Fig. 1a in Maheswaranathan et al., 2019) describing the task visually.
>
> > When trying to understand the ring attractor I eventually decided that r and s in equation 5 were phi(g) and g in equation 25...
>
> This is correct and we apologize for the oversight. Thank you for catching the error.
>
> > Further, in appendix 9 you say "in our deformation analysis we only changed phi after simulation, which does not affect the ring topology but modifies the geometry”...
>
> We apologize for the lack of clarity and will  reword this. In our experiment, we applied a new nonlinearity to the synaptic activations to get a different output of the rates of the neurons. So we transformed the data on which we applied DSA, while still using the same underlying trials--you can also think of this as a modification of how we observed the data. The goal was to demonstrate invariance to smooth deformations. In Fig. 4b, the Procrustes distance increases with the magnitude of the nonlinearity, which verifies that the data is deforming . We hope this makes sense and are happy to clarify further.
>
> > The claim that DSA measures topological structure of the dynamics would be fairly remarkable... but is not proven...
>
> We thank the reviewer for this important point and have since identified a proof that we will include. We can also add fudge words as you suggested. The proof is simple and we will include the formal proof in the paper: For topologically conjugate systems related by a diffeomorphism, their Koopman operators have the *exact same* eigenvalues, and the eigenfunctions are related by the diffeomorphism (Proposition 7, Budisic et al. 2012). This implies that the DMDs--the finite approximation of the Koopman operator--are related via the similarity transform.
>
> > It seems like the scale of the MDS dimensions is argued to be meaningful... what does it correspond to?
>
> As you identified, the dimensions are linear projections of the data, so a small scale implies small dissimilarity between systems. Thus in Fig 3b, being clustered around 0 means that the RDM is made up of mostly zeros. If they were scattered farther apart it would suggest that the models could be very different from one another. In revisions, we can display some statistics over the RDM or explain this logic in more detail.
>
> > How well do the linearised dynamics capture the underlying network dynamics?...
>
> In Appendix 8, we swept a range of DMD hyperparameters and found that the DMD fit the flip-flop system relatively well. Nevertheless, you are correct that most nonlinear systems are not linearizable and the DMD cannot perfectly fit. *However*, based on our analysis, that is not a problem for DSA, as the relevant outcome is for the system to converge to an approximation of the Koopman operator. Thus even if the dynamics are not perfectly fit, the similarity transform distance will identify conjugate systems as similar.
>
> We should clarify that forcing is used to generate *autonomous* nonlinear dynamics, in that the system can be fed its own predictions and thus reproduce the dynamics. Time-delay observables have been used to approximate of the Koopman operator without forcing in several settings (Arbabi & Mezić 2016, Brunton et. al. 2016) and thus the omission of the forcing term should not (and indeed does not) impact DSA.
>
> > I did not buy one claim, could you elaborate:...
>
> We thank the reviewer for this concern and very much agree. Here's what we meant: in a high-dimensional system that has low dimensional dynamics (for example, 99% of the variance is explained by the first 5 dimensions), we can effectively project to that 5-d subspace and the system will evolve in the same way as in the ambient space. This is because the dynamics components orthogonal to those dimensions is negligible. But in general this is not true, as in your example. We can delete this line if it would eliminate confusion, or can add a section in the appendix about the viability of PCA in DSA in certain conditions (manifold hypothesis).
>
> > You claim that all networks that solve the 3-bit flip-flop task have the same 8 fixed point structure, line 219, do you have any evidence for that?
>
> Yes, this was shown in Maheswaranthan et al., 2019 and Sussillo and Barak 2013--they perform an extensive numerical analysis. We apologize for the oversight and thank you for the suggestion -  we will cite directly after making this claim.
>
> > Why did you not choose figure 3 to also back up your claim...
>
> We apologize for this--it was a semantic distinction ('similarity' versus dissimilarity). Nevertheless, we should connect the concept of capturing topological similarity to Fig. 3.
>
> > Related to the topological claim, my intuition must be off...
>
> This is an interesting scenario, and is related to Fig. 4a. In DSA, we compare the two systems on the eigen-time-delay coordinates. In doing so, singular values are factored out, which "re-circularizes" your ellipse. In Williams et al. (2021), they solve a similar problem by whitening the data, and fitting HAVOK in the eigen-time-delay coordinates is equivalent to PCA whitening on the data.

---

> > ### Comment · Reviewer_TLG4 · 2023-08-13
> > **Reviewer Response**
> >
> > I thank the authors for their thorough response. Some comments:
> >
> > First, regarding the nonlinearity you applied to the ring attractor network, I think I understand now that you ran ring attractor dynamics, got a set of firing rates through time, then mapped those through some nonlinearity to create a morphed version of the same ring as shown in figure 4. If so that now makes sense to me but could definitely be more clearly explained. Further I liked another reviewer's worry that the DSA does not equal 0, and that you have worked out a way of making it equal 0 (though it does bring concerns about how hard the transformation is to optimise).
> >
> > Second, I'm glad to hear you have a proof that your method extracts the topological structure, that's very cool. If it is correct then there's no need for including the fudge words! (Unless I'm misunderstanding what the fudge words are for, surely you want to make the stronger claim if you have the theoretical work to support it?)
> >
> > I did not understand the author's explanation about why the forcing term doesn't matter. I think I get that maybe you only care about the internal dynamics operator (the Koopman operator) and perhaps it is true that with or without forcing you'd expect to extract the same Koopman operator, is that what you are saying? To the statement you make about autonomous nonlinear dyanmics in your response, is that not what you are doing?
> >
> > I accept your argument that if all the activity is in a small set of PCs then the dynamics within that space should be the same as the whole system, and I would indeed be satisfied if you either deleted that claim or qualified it, as you suggest.
> >
> > Finally, this is more for my own curiosity, but could you explain what you mean by 'singular values are factored out' in your last piece of response? My intuition is still off! It seems like the example I presented, the ellipse (and you are right, this is basically figure 4c), was wrong because the time lagging is key, so I tried thinking about the effect of time lagging. What I don't get is let's say I create a load of time-lagged versions of the same co-ordinates and then do eigendecomposition, then I would expect the areas of space in which the activity spends more time to be over-represented, upregulating their weight in the eigendecomposition. I think this would be the furthest points of the ellipse, which are those along which most of the data variance lies anyway, so it seems like lagging only make the situation worse?! Does this argument make sense? Do you know what's going wrong?

---

> > > ### Author Response · Authors · 2023-08-15
> > > **Response to Reviewer TLG4's Response**
> > >
> > > Thank you for your comments, and we are glad to be able to have satisfied some of your concerns. We will certainly implement your feedback in our revisions. Here are the answers to your outstanding questions:
> > >
> > > >  perhaps it is true that with or without forcing you'd expect to extract the same Koopman operator, is that what you are saying?
> > >
> > > This is exactly correct, the same model fitting algorithm (reduced-rank regression) is used in each scenario. The forcing term is chosen after fitting the HAVOK model, by selecting the last r bases of the regression model. That is, the DMD matrix with $n$ modes and $r$ forcing terms is identified by learning a DMD matrix with $n+r$ modes. This means that fitting a HAVOK model with $n$ modes and no forcing would learn the same DMD matrix.
> > >
> > > > To the statement you make about autonomous nonlinear dynamics in your response, is that not what you are doing?
> > >
> > > In our case we are using HAVOK for its statistical properties in identifying the DMD matrix, not for its generative properties. We can illustrate what we mean by generating autonomous nonlinear dynamics with an example: Consider the HAVOK model fit to the Lorenz attractor--in the non-forced DMD setting, running this model autonomously (i.e. evolving the HAVOK model via tail-biting  - generating predictions recursively based on prior predictions) cannot regenerate the full chaotic dynamics. This is because it is a linear model. Here we mean it does not regenerate the full dynamics in that the model cannot switch lobes of the attractor. The forcing term can be used to remedy this issue and allow for autonomous chaotic dynamics generated by the tail-biting procedure.
> > >
> > > > Finally, this is more for my own curiosity, but could you explain what you mean by 'singular values are factored out' in your last piece of response?
> > >
> > > Of course. Let's consider how this relates to PCA whitening (and we will add this to the supplementary information, as this question is very important and worth spending more time explaining). In PCA whitening, we are able to normalize the covariance matrix of the data:
> > >
> > > $HH^T = U\Sigma^2U^{-1}$, where $H$ is the (already-centered) data matrix (Hankel Matrix), $U$ is the eigenvector matrix of the covariance matrix, and $\Sigma^2$ are the eigenvalues, which are equivalent to the squared singular values of the data. Then the PCA whitening on our data H is:
> > > $H \leftarrow \Sigma^{-1}U^{-1}H$
> > >
> > > Now, consider $H = U\Sigma V^T$. PCA whitening mathematically amounts to the following expression:
> > > $H \leftarrow \Sigma^{-1}U^{-1}U\Sigma V^T = V^T$ as we cancel terms.
> > >
> > > Thus PCA whitening is equivalent to simply extracting the right singular vectors from the SVD, which is exactly the transformation of the data on which we fit our DMD matrices! Thus we are able to 'sphere' our data just like in PCA whitening. Here, sphering is what is meant by "factoring out the singular values". That being said, we agree that we need to clarify what we mean by "factoring out the singular values" and will substitute this explanation instead. Thank you for your comment.

---

> > > > ### Comment · Reviewer_TLG4 · 2023-08-18
> > > > **Reviewer Response II**
> > > >
> > > > Thank you for these clarifications, they have certainly helped me settle this knowledge in my mind!

---

### Official Review · Reviewer_mRC1 · 2023-07-04

**Soundness:** 3 good
**Presentation:** 3 good
**Contribution:** 3 good
**Rating:** 5
**Confidence:** 3

**Summary:**

Understanding how different neural populations process a particular computation is critical for the study of brain computation, the development of brain-inspired technologies like brain-computer interfaces (BCI), and AI applications. Existing methods compare the underlying representations based on the spatial geometry of the latents. However, they fail to capture the dynamics, which are believed to drive the specific computation. To overcome this limitation, this work introduces a new analysis method that first extracts the dynamical structure of the neural network and then aligns these representations for comparison. They test the model when applied to several case studies training RNNs across different tasks and architectures. The proposed analysis can capture similarities between the computations across different networks while other models fail.

**Strengths:**

The paper is clearly presented and technically sound. The method was tested and shown to work well when applied to artificial neural networks trained to solve different tasks and architectures, resulting in different dynamics and geometries. The analysis expands on two existing methods and combines them to understand the comparison between neural dynamics, which support the computation. Understanding similarities and differences between neural networks solving similar tasks is critical for the fields of neuroscience and AI. In this work, they showed how this analysis can identify overlap between computation solutions when other alternative methods fail. Moreover, the authors suggest that it could be used to understand biological network dynamics.

**Weaknesses:**

While the authors provide compelling results supporting the relevance of this metric when applied to RNNs, they have not applied it to neural data. Since this project is largely motivated by the potential study of biological neural circuits and the application to BCIs, applying the analysis to biological data is critical to fully grasp the scope of applications of the proposed method and therefore its significance. As mentioned in the manuscript, the analysis is "[...] useful for experimental neuroscientists due to its simplicity of implementation and ease of interpretation due to its linearity." However, while its linearity provides the aforementioned benefits, it also imposes some limitations that could be interesting to explore. Lastly, when it comes to the application of alignment for BCI applications, it is important to note that the reported training time is between one hour and one day, making it impossible to use in its current state for practical applications when daily alignment is presumably needed. Along the same line of thought, it would be interesting to explore and note the data demands for the new method to work robustly.

**Questions:**

The case studies using RNNs show the strengths of the proposed analysis in comparison to other methods. However, it is not always the case that the new analysis is superior to the alternatives. It would be interesting to explore cases when it isn't and why that is the case. As described, the modified Procrustes alignment seems to only support pairwise comparisons. If that is the case, are there limitations when comparing multiple networks? Does one have to set a reference network to compare all the others?

**Limitations:**

The authors thoroughly address the limitations on hyperparameter training and computational cost.

---

> ### Author Rebuttal · Authors · 2023-08-08
>
> We thank the reviewer for their feedback and greatly appreciate that they think the paper is clear and technically sound. We hope that our response will clarify some of the weaknesses and questions that the reviewer mentioned:
>
> > Applications to neural data
>
> While we haven't applied to neural data, we should note that the first example in the disentangling learning rules section (Fig 5a) utilized data from large-scale CNNs, trained on Imagenet (for example, a deep resnet, Nayebi et al., 2020). Although even this isn't quite as complex as biological neural data, we hope that this setting points to DSA's viability on neural data as well. We also agree that the application to neural data is one of the most exciting and important applications of our method, and are currently developing follow up work involving neural data.
>
> > limitations of the linear method
>
> While the model itself is linear, we note that the DSA gains nonlinear power via the delay embedding, which acts similar to a kernel function in an SVM. However, it is still worth thinking about DSA's limitations, and we're actively working on a deeper theoretical analysis.
>
> > BCI limitations and training time
>
> We apologize for the lack of clarity in our reported training time--we should note that the time to fit *one* DSA comparison is on the order of a minute or less, provided the number of optimization iterations is not absurdly large. We noted in the limitation that the total time per experiment (i.e., a figure) was between one hour and one day because we were performing tens of thousands of comparisons. For example, in Figure 2, we compared 240 neural networks pairwise, or around 29 thousand comparisons in total. Naturally, this takes a long time to compute, but each individual comparison is relatively fast, and we have GPU capability which drastically improves performance. We can clarify this in the limitations, and we hope that this explains what we meant by one hour to one day of computation.
>
> > data demands
>
> We agree that the question of how our method scales is a very interesting one to consider. We believe that the key limiting factor is in the DMD, which should scale with respect to the underlying dimensionality of the data manifold. We are actively studying this in a followup project--thank you for your suggestion.
>
> > superiority of DSA vs alternatives
>
> We agree that DSA is not always superior. We wish to emphasize that we do not intend to replace shape metrics, but rather to supplement them with a new comparison method to be done in parallel. Geometry is of course an important aspect to consider, and there exist many use cases for geometric methods. The goal of our paper was to highlight that if one cares about dynamics, there are situations in which geometric methods will not suffice to capture relevant similarities or dissimilarities at the level of the underlying dynamics.
>
> > Pairwise comparisons in DSA
>
> This is a similar concern for standard shape metrics--by virtue of being distance metrics, they can only compare pairwise. When comparing multiple networks (which we did in a few settings) there are a couple clear options, as you've mentioned: (1) compare pairwise, which results in a representational dissimilarity matrix of $R^{n \times n}$ (Fig 2,3,5) or (2) we can fix a reference network (Fig 4) for a representational dissimilarity matrix of $R^{n \times 1}$. In our open source code (which we are releasing), we've made it easy to do either one, with the option of setting hyperparameters for each model individually or in aggregate.

---

> > ### Comment · Reviewer_mRC1 · 2023-08-15
> >
> > i thank the authors for their through and detail answers. I believe that there is a lot of potential for this method. Specially since the computational demand is not that high. Still, in order to demonstrate the true value of this approach it would have to be tested on neural data. I am looking forward to your future work.

---

### Official Review · Reviewer_cz18 · 2023-07-06

**Soundness:** 4 excellent
**Presentation:** 4 excellent
**Contribution:** 4 excellent
**Rating:** 8
**Confidence:** 4

**Summary:**

The authors developed a new computational tool named Dynamical Similarity Analysis (DSA) to measure the similarity between two systems focusing on the dynamics. They constructed the method by combining and modifying Dynamical Mode Decomposition and Statistical Shape Analysis. Their method successfully identified dynamical similarities between systems with different geometries, and conversely distinguished systems with different dynamics with similar geometries. They also used their method to distinguish learning rules underlying measured neural dynamics.

**Strengths:**

I enjoyed this paper, which I found to be both elegant and useful. This work is a novel combination of previously developed techniques. This work is significant because the standard approach for comparing recurrent neural networks merely focuses on the spatial similarities of latent states, ignoring the importance of the difference between temporal similarities. Their proposal avoids the limitations of traditional tools by accounting for neural dynamics. Overall, the authors have clearly delivered their statement, proposed method, experiment, and results. The method is well-grounded and the ideas behind the proposed technique are well-explained and justified. The writing was clear, and the figures were largely appealing (although some of the labels became small and/or blurry). They analyzed when their method is superior to traditional methods, and used simple tasks to demonstrate this superiority. The distinguishing of learning rules from data was particularly nice.

**Weaknesses:**

The authors could better explain and motivate the conventional spatial Procrustes Analysis, as that would help some readers take the next step for the temporal version. They could clarify the interpretation of their similarity transform CAC^-1, as I felt that the explanation that C^-1 “rotates [the vectors’] positions as well” was not a great explanation.

Minor: It’s strange to talk favorably in Figure 2 about DSA giving better accuracy but having lower accuracy scores, when discriminating between identical computations. If pairs of networks are correctly identified as “the same dynamics” then NOT distinguishing the networks should have higher accuracy. You might want “discriminability” on the vertical axis of Figure 2d.

**Questions:**

None. Very nice work.

**Limitations:**

The authors adequately addressed limitations.

---

> ### Author Rebuttal · Authors · 2023-08-08
>
> We thank the reviewer for their feedback and are very happy that they found the paper elegant, significant, and useful! We found their suggestions for improvements quite helpful as well, and hope that we can implement them in a manner that they agree with:
>
> In their point that some of the figure's labels became small and blurry, we think they are referring to Figure 1? If so, we uploaded a figure that was too large and have since compressed it--we believe that will solve this problem. We are also happy to increase label sizes as needed.
>
> We also agree that we did not explain and motivate Procrustes well, instead skipping over it to motivate our new metric. In the main paper, we can add an intuitive explanation of Procrustes describing that it captures distance in a manner that is invariant to orthogonal transformations, which is relevant because the coordinates of the space are not important for the setting of comparing neural representations.
>
> Regarding the interpretation of $CAC^{-1}$ , we believe that the quoted interpretation of the similarity transform goes very well with our Figure 1 in the Appendix, and can move that explanation there so that they align. We can replace the interpretation in the main paper with a more mathematically motivated one based on homomeorphisms  between conjugate dynamical systems, in this case linear systems, or another alternative interpretation that is clearer.
>
> Finally, we agree that our description of "good" and "bad" in Figure 2 does contradict standard perceptions with respect to classification accuracy, and would appreciate your feedback on the rewording. We suggest that we could change the phrase "the networks intermingle" in L231 to "the networks are indiscriminable", and then reference Fig 2d in that line. We can chance the vertical axis to "discriminability" as you suggested, and add one more sentence at the end of section 1 explaining that it is desirable to our metric to have discriminability close to chance in this example. Alternatively, we could define a metric "indiscriminability" as 1-test\_acc, but this may be more confusing?

---

### Official Review · Reviewer_f1wF · 2023-07-07

**Soundness:** 3 good
**Presentation:** 3 good
**Contribution:** 2 fair
**Rating:** 6
**Confidence:** 5

**Summary:**

The authors propose Dynamical Similarity Analysis (DSA), a method for assessing the dynamical similarity between dynamical systems. The method combines ideas from the data-driven Koopman operator literature and a Procrustes-type distance between linear operators. By focusing on dynamics rather than geometry, the method is able to reveal the similarities between RNNs of different architectures trained to perform the same tasks and distinguish between dynamical systems with similar time-averaged states but different underlying dynamics.

**Strengths:**

1. The method is simple and appealing. Appealing use of time delay embeddings -- transform the temporal task into a geometric task.
2. DSA works very well on most of the experiments: Figures 1, 2, 3, and 5.

**Weaknesses:**

1. Should compare to baseline spectral method, for example PCs of CPSD matrix...
2. Figure 4: DSA is not zero. l262 speculates about numerical approximation error, but this seems to be fairly large. Ideally, the line for DSA would stay very close to zero. If this is not due to numerical approximation error, should this be taken as a failure case of DSA? The text mentions the low variability of the DSA distance across values of beta and C, but it seems like the magnitude is more relevant in this example.
3. The manuscript would improve with better theoretical grounding. For example, something in the vein of what is proposed in the paragraph starting at line 317 would greatly improve the paper.

**Questions:**

1. Figure 1 uses angular distance -- where is this defined?
2. There are several comments made about noise in the manuscript. For example, l58: "DSA factors noise into its estimate of neural dynamics." Also l172. I remain confused about how noise interacts with different aspects of the model. For example, how does Takens' theorem interact with noise?
3. How are lags and rank chosen for each experiment. Are the results robust to these hyperparameters? They seem to vary widely from experiment to experiment.

Suggestions/comments:
* Define what shape metrics are, or perhaps explicitly define one common one.
* Condition is not defined in the setup.
* Quantifying dissimilarity: Figure 3: MDS is a good visualization, but there should also be predictions as in Figure 2 or distances as in Figure 1. This should be uniform across figures.

**Limitations:**

* Paragraph starting at l333: I would expect one would need to fit a model that fits task-relevant dynamics. That is, DSA would probably require some modifications before it is applied to neural data.

---

> ### Author Rebuttal · Authors · 2023-08-08
>
> We thank the reviewer for their helpful feedback and suggestions. We are glad that the simplicity of the method was appealing, and we hope that our responses to each of your comments will allow you to further appreciate DSA.
>
> We appreciate the suggestion to add a spectral method of comparison, and agree that it would be valuable to compare to additional baseline metrics. After some investigation, though, we have to admit we are unclear on what the process would be for the baseline method you suggested. Would it be possible for you to elaborate on your suggestion by pointing us to a specific reference? This would help us incorporate your feedback better.
>
> > Figure 4: DSA is not zero.
>
> We agree that this is a problem, and we are happy to share that we have solved it since submitting the paper. We will update the figure accordingly--now, DSA reports \~0.06 with a standard deviation of 0.01 across the deformation values (Fig 4) . The error was in fact numerical approximation error due to optimization, and we solved it by including a multilayer perceptron in our pipeline, which makes the loss landscape more amenable to gradient descent. In our open source code (to-be-released accompanying the paper), we implemented a unit test asserting that a matrix $A$ and its similarity transform $CAC^{-1}$ would be identified as similar up to a threshold, which now passes.
>
> > The manuscript would improve with better theoretical grounding.
>
> We agree with the reviewer on this weakness and are happy to share that we have identified theoretical grounding. The proof is quite straightforward and we can describe it in words here, but we will certainly include the formal proof in the paper: For two topologically conjugate systems $f(x),g(y)$ related by a diffeomorphism $y = \phi(x)$ , their Koopman operators $K_x,K_y$ have the \textit{exact same} eigenvalues, and the eigenfunctions are related by the same diffeomorphism $\phi$ (Proposition 7, Budisic et al. 2012, "Applied Koopmanism"). This implies that the DMDs - the finite approximation of the Koopman operator - are related via the similarity transform, as similar matrices have identical eigenvalues.  Thus, we can identify topological conjugacy of two nonlinear dynamical systems with DSA.
>
> > Figure 1 uses angular distances -- where is this defined?
>
> We only defined this in the appendix, section 3 (eq. 15). We agree that this may be confusing and are happy to add it into the methods section.
>
> > comments made about noise in the manuscript
>
> We agree that our description here was confusing, and are happy to clarify both here and in the paper.
> Noise (in particular, small noise perturbations) is important for estimating the DMD appropriately by reducing degeneracy in the dynamics matrix estimation. As a motivating example, consider estimating the dynamics of a system observed close to a fixed point. Without noise, the regression would be degenerate as the system is barely changing. But with small noise perturbations, there is enough variance in the observed data to enable the construction of an appropriate dynamics matrix.  A more complex argument for the importance of noise is also described by Galgali et al., Nat. Neuro (2023), from which we identified the examples in Fig. 3.  Essentially, without small noise perturbations, our method would not be able to distinguish the examples in Fig 3, as the condition-averaged dynamics are exactly equivalent.
>
> The reviewer also brings up an interesting question about how noise interacts of theoretical backing for our method, such as Takens' theorem. In the specific case of Taken's theorem, it seems as though there have been attempts to rigorously characterize the quality of the state space reconstruction at varying signal to noise ratios (see Casdagli et. al. 1991). However, again noting that our description was confusing, we wish to clarify that we meant to claim not that our method is robust to large amounts of noise, but rather that the method is able to harness small noise perturbations in order to obtain appropriate models of the dynamics.
>
>
> > How are lags and rank chosen for experiment
>
> Like most DMD papers, we chose them heuristically, based on our knowledge of the dimensionality of the system (both in terms of intrinsic manifold and number of neurons), and based on how well the DMD model predicted the next step data (3 metrics here, MSE, R\^2 and Correlation). However, the results are robust to hyperparameters--in Appendix Fig section 11 we showed that the predictivity metrics are optimized across a wide range of ranks and lags, which are reasonable proxies for how well DSA will do. We are aware of algorithms that optimize these hyperparameters based on predictivity metrics (Ahamed et al., Nature Physics 2020) but chose not to use them here, as picking hyperparameters by hand was surprisingly easy to get our method to work.
>
> > Suggestions
>
> Regarding your suggestions, we agree with all of them. We defined the Procrustes Analysis (a particular shape metric, William et al., 2021) in equation 3 but realize that we did not explain its function very well, so we will elaborate upon it in our revisions. Specifically, we plan to add an intuitive explanation of Procrustes describing that it captures distance in a manner that is invariant to orthogonal transformations, which is relevant because the coordinates of the space are not important for the setting of comparing neural representations. We also agree that we need to define what we mean by condition, and will do this in the revision--a condition in systems neuroscience experiments is the context that the experimenter varies (input stimuli and desired output mapping, e.g. the variable input drive in Fig 3--one condition has input $u$ with positive magnitude, the other condition has negative input). Finally, we agree that we should quantify the discriminability achieved by DSA in a panel in Fig. 3 and will add that into the paper.

---

> > ### Comment · Reviewer_f1wF · 2023-08-20
> >
> > **Comparing to a spectral method:** I apologize to the authors for how vague my suggestion was in the original review. My concern was that the proposed method uses time information via DMD while the competing approach (Williams et al. 2022) does not use time information. I recognize that part of the contribution of this work is to extend the general sort of analysis in Williams et al. (2022) to the time domain and agree that this a valuable contribution. However, apart from this aspect of the experiment, the comparisons presented in Figures 2b, 3b, and others is possibly misleading/unfair because the competing method do not have access to any time information. This leads me to ask: could there be a stronger baseline which i) does have access to time information and ii) uses existing approaches? In my opinion, this would provide more relevant context for interpreting the experimental results.
> >
> > Here is one concrete proposal that fits these two criteria: apply the method of Williams et al. (2022) to a spectral representation of the data. In more detail, given a single timeseries $X \in \mathbb{R}^{n \times t}$, estimate the cross power spectral density (for example, using ``scipy.signal.csd`` and taking the absolute values, discarding phase information) to create $\tilde{X} \in \mathbb{R}^{n \times n \times f}$ where $f$ denotes the number of frequency bins. Then simply treat these as features to input to the Procrustes metric (Eq. 4).
> >
> > **Theoretical grounding:** I am glad that the authors have found a more explicit theoretical grounding for their method, which will substantially improve the paper.
> >
> > **Figure 4:** I am glad that the issue in Figure 4 has been fixed.
> >
> > **Noise:** I thank the authors for pointing out Casdagli et al. (1991). Is it fair to summarize the noise situation, meaning nondeterministic dynamics, as follows? i) The theory appealed to in motivating the method (Taken's theorem) was developed for deterministic settings. ii) There have been attempts to systematically study the effects of noise (eg Casdagli et al., 1991) and it has been found that noise somewhat complicates the task of state space reconstruction. iii) Small amounts of noise are helpful in practice for estimating the dynamics matrices and distinguishing different dynamics (as in Figure 3). Is this a fair characterization, I suggest mentioning all three points explicitly in the manuscript.
> >
> > **Matrix similarity:**
> > The orthogonal group of matrices consists of two connected components, but the Cayley transform used (Eq. 5 in the supplementary material) produces only matrices in the special orthogonal group. Due to this, it seems that references to $O(n)$ in the text should be replaced with $SO(n)$.
> >
> > A minor suggestion: it may be helpful for clarity to point out in the text around Eq. 4 that $d=0$ implies the two $A$ matrices are (unitarily) similar. It would also be helpful to mention in the main text that the converse is not true in general, due to the restriction of $C$ to $SO(n)$.
> >
> > I'm curious if the authors tried optimizing the generic matrix similarity metric (Eq. 4 in the supplement) using gradient descent. Does the nonconvexity pose a practical problem?
> >
> > On a related note, it may be safe to assume that both $A$ matrices contain distinct eigenvalues, given that they are estimated using noisy experimental data. If this is the case, it may be possible to define an analogous metric to DSA (call it DSA') by computing a metric on the eigenvalues themselves, for example a Wasserstein distance between the two sets of eigenvalues. This would have the advantage of the following implication: if the eigenvalues of the A matrices are distinct and DSA' is 0, then the A matrices are similar. Note that this is slightly more general than what can be said about the existing DSA. Of course DSA' also depends practically on the numerical stability of the eigenvalue estimation.
> >
> > **One more minor comment:** It seems like DSA is not invariant to changes in timescales. For example, if we have two identical systems with one evolving a few percent faster than the other, the DSA score will be nonzero. This could be worth mentioning as a limitation, given that we may not expect neural dynamics to operate at identical speeds on different trials or across different animals.
> >
> > Thank you to the authors for your detailed response. Most of my concerns have been adequately addressed, and so I will raise my score. However, I still think the manuscript would be substantially improved with some comparison to a spectral method like the one described above. Apologies for my slow response.

---

> > > ### Author Response · Authors · 2023-08-21
> > >
> > > We thank the reviewer for their extensive comments, and greatly appreciate their raising of the score. We are glad to have addressed your concerns, and believe that the points you have raised in this response are quite valuable.
> > >
> > > **Comparing to a spectral method**: We believe that we understand the proposed spectral method, and agree that it serves as another relevant baseline to cover both the purely spatial and the purely temporal view of dynamics.
> > >
> > > **Noise**: Yes, your points are well-described and we will add them to the manuscript.
> > >
> > > **Matrix similarity**: You are correct that the Cayley Map restricts to $SO(n)$, and we will change the text accordingly. Likewise, you are correct that the metrized version of the task only identifies similarity up to unitary transforms, and we need to mention that as well. We also agree that testing the general metric ($C \in GL(n)$) is highly relevant and would capture general topological equivalence, even if it is not a proper metric, as far as we know. We will mention this and describe the relevant cases for using each group in DSA.
> > >
> > > We previously considered an idea similar to your DSA', but found that just comparing the eigenvalues will not work in cases of non-normality. Consider this simple example in which case the pure eigenvalue comparison will fail: https://math.stackexchange.com/questions/1955796/find-two-2-times2-matrices-which-have-the-same-eigenvalues-but-are-not-simila
> > >
> > > Thank you again for your insightful and thorough comments.

---

### Decision · Program_Chairs · 2023-09-21

**Decision:**

Accept (poster)

**Comment:**

The authors introduce Dynamical Similarity Analysis (DSA), a novel method for the dynamics of neural networks and other dynamical systems. By combining advanced techniques like the data-driven Koopman operator and Procrustes Analysis, DSA effectively captures dynamical similarities in various architectures and can distinguish different underlying dynamics, offering a robust tool for understanding neural computations and testing recurrent neural network models. This is important work that will be of broad interest to NeurIPS audience and beyond.